# Activation dynamics and assembly of root zone soil bacterial communities in response to stress-associated phytohormones

Sreejata Bandopadhyay,[1,2] Oishi Bagchi,[1,2] Ashley Shade[3]

ABSTRACT  Plants can "cry for help" to recruit supportive microbiome members during stress, but the precise signals a plant uses to activate and assemble these microorganisms remain unclear. We evaluated the activation dynamics of root zone soil bacteria in response to phytohormones produced when plants are stressed, hypothesizing that responsive taxa could support plant resilience. We conducted a mesocosm experiment using root zone soil collected from the planted fields of two crops: the annual legume common bean (*Phaseolus vulgaris* L.) and the perennial grass switchgrass (*Panicum virgatum*). In the absence of any plant, we inactivated the root zone microbiome by drying the soil and then added abscisic acid, salicylic acid, a carrier control, or water to test their capacities to reactivate microbiome members and assessed responses for 2 weeks. Using amplicon sequencing of the 16S rRNA and rRNA genes to determine active populations, we found several actinobacteria that became active after exposure to abscisic acid and salicylic acid, with *Microbispora* lineages being especially responsive. While some taxa activated only in one crop's root soil, others were activated in both crops' soils in response to the same phytohormone. By comparing microbes that immediately activated 24 h after phytohormone addition with those that activated and also persisted over several days, we distinguished taxa that responded to phytohormones as signals from those that potentially also used them as resources. This work suggests that different root zone bacteria exhibit distinct specificities to phytohormones, providing insights into the signals by which plants may "cry for help" to recruit bacteria.

IMPORTANCE Global food security is a pressing societal challenge that has been exacerbated by climate change and other anthropogenic stressors on the environment. Microbial bioinoculants are a promising solution for improving crop health and resilience, but ensuring their persistence and activation in the field remains a significant challenge. This study examined how dormant root-zone-associated bacteria reactivate after exposure to the plant stress hormones abscisic acid and salicylic acid. The experiment revealed that certain bacterial taxa could reactivate in response to these plant stress signals and persist for at least 2 weeks. This work advances our understanding of the potential cues for reactivating beneficial plant-associated microbes and supports the goal of developing microbial solutions for sustainable agriculture.

KEYWORDS  plant microbiome, rhizosphere, salicylic acid, abscisic acid, mesocosm, 16S ratios, amplicon sequencing, transcriptionally active populations, reactivation, before-after-control-impact design (BACI)

Plant hormones, called *phytohormones*, are essential for a plant's response to stress. Phytohormones improve germination, upregulate antioxidant systems to reduce reactive oxygen species (ROS) accumulation, induce systemic acquired resistance, and assist in cell signaling. They are also vital growth regulators that can impact plant metabolism and are often externally applied to plants to improve plant growth under

Editor Blaire Steven, Connecticut Agricultural Experiment Station, New Haven, Connecticut, USA

Address correspondence to Ashley Shade, ashley.shade@cnrs.fr.

The authors declare no conflict of interest.

See the funding table on p. 15.

stress conditions (1). Several microbes are known to produce phytohormones or phytohormone mimics (2–5) which influence phytohormone networks in plants and subsequently affect microbial metabolism. Thus, phytohormone regulation in plants is considered to be connected to microbial phytohormone production, together forming inter-organismal systems that can support plants (6).

The plant rhizosphere and proximal root zone are nutrient-rich compartments due to root exudation and, thus, are heavily colonized by microbes (7, 8). Sugars and amino acids within root exudates serve as nutrients for microbes (9, 10), which, in turn, can assist plants by improving their nutrient acquisition (11) and protecting against stress (12, 13). Specifically, microbial production of phytohormones (6) such as abscisic acid (ABA), auxins (indole 3-acetic acid), salicylic acid (SA), and cytokinins can improve a plant's ability to cope with stress. These well-known plant-microbe interactions have been extensively investigated (6, 14–16).

The phytohormones ABA and SA play an important role in plant abiotic stress tolerance. Abscisic acid belongs to a group of phytohormones called sesquiterpenoids, which regulate plant growth. ABA-induced signaling can regulate the expression of stress-responsive genes and contribute to stress tolerance (17). During drought, ABA induces stomatal closure, thereby controlling transpiration (18). Exogenous ABA application to plant seeds before sowing can protect a drought-sensitive wheat cultivar from drought-induced oxidative damage by increasing the activity of the antioxidant enzyme peroxidase (19). Salicylic acid also modulates the activities of antioxidant enzymes (20, 21). Similar to ABA, the application of SA to plant tissues has been shown to relieve water stress (22), potentially enabling continued plant growth despite drought (21).

While much is understood about a plant's coping mechanisms during stress via phytohormone regulation of cellular pathways, we are still learning how phytohormones impact rhizosphere microbiome assembly during stress. It has been demonstrated that SA signaling can enhance microbial community diversity and facilitate the colonization of specific taxa (23–25). Similarly, low-molecular-weight organic acids, such as malic acid, citric acid, and fumaric acid, secreted by roots have also been shown to serve as a source of carbon substrate and a signaling molecule for rhizobacteria recruitment (26–28). Direct evidence of phytohormones serving as a carbon source for microbes is more limited; however, some studies demonstrated increased soil respiration when treated with one mM abscisic acid (29) and higher soil respiration when soils were treated with root exudates from drought-exposed plants compared to exudates from non-droughted plants (30). However, there is indirect evidence that rhizobacteria recruitment can be enhanced in response to specific phytohormone precursors such as L-tryptophan, a precursor for auxin synthesis (31). This study showed that auxin-producing *Pseudomonas fluorescens* stimulated the root growth of radish, which exudes high levels of tryptophan. If microbes use phytohormones as a carbon source, it is possible that they could cause a soil-priming effect that stimulates microbial respiration. This has been shown in experiments where low levels of jasmonic acid (JA) and 1-aminocyclopropane-1-carboxylic acid (ACC) addition to soils caused a relatively large increase in soil respiration (29). Such priming effects have been documented for various organic substrates in root exudates (32–35).

The rhizosphere microbiome members recruited during stress are expected to support a plant's ability to recover from that stress or exploit the plant during stress. Thus, understanding the role of stress-associated phytohormones in determining rhizosphere microbiome assembly is important for fully understanding the complex dialog and feedback between a plant and its microbiome. Phytohormone release or leakage into the root zone is also one potential mechanism by which plants may "cry for help" via root exudation to attract supportive microbial partners from the root zone to the rhizosphere (36).

A complicating factor in understanding plant-soil-microbial interactions during stress is the unclear contribution of those soil microbes that reactivate from dormancy. A large

fraction of the soil bacterial community can be dormant at a given time (80%–90% of cells and 55% of taxa), with dormant cells remaining inactive for prolonged periods (37). Even nutrient-rich rhizosphere soil can have substantial microbial dormancy, with around 40%-80% inactive cells reported in the rhizosphere soil of bean plants (38). Thus, we reasoned that the rhizosphere and proximal root-zone compartment may be highly dynamic in activity switching during stress, with some taxa reactivating in response to plant signals (and possibly supporting the plant's stress response). In contrast, others initiate dormancy to avoid the stress by protecting themselves (39).

The overarching objective of this study was to understand how the root-zone bacterial microbiome responds to phytohormones that signal plant stress, both in terms of activation and inactivation. Our null hypothesis was that the microbiome does not react to phytohormone exposure. We had three alternate and non-mutually exclusive hypotheses:

> **H1: The microbiomes of root-zone soils from different plants have different activation and assembly dynamics in response to phytohormones.** The motivation for this hypothesis stems from our previous work on these plant genotypes (40) and the literature, which indicates that different plant species harbor distinct bacterial microbiome structures. If supported, this hypothesis suggests the importance of the plant's ability to locally assemble and activate "custom" microbiome members to respond to its own phytohormones.

> **H2: Certain bacterial lineages generally activate in response to phytohormones regardless of the plant.** The rationale for this hypothesis is that the phytohormones could serve as resources for microbes. Therefore, the bacteria that can utilize exogenous phytohormones may generally activate if provided either SA or ABA from any plant species. This activation pattern may indicate that taxa respond to resource availability, not a stress-specific "cry for help."

> **H3: There is a hormone-specific activation of certain bacterial taxa.** Each phytohormone indicates a different type of stress for the plant: mainly pathogen infection in the case of SA and drought in the case of ABA. Thus, if some bacterial lineages activate after exposure to one of these phytohormones but not both, it would be consistent with a bacterial taxon's response to a plant's "cry for help" given a specific stress.

We addressed these hypotheses using a mesocosm experiment in which we exposed root-zone soils to either a stress-associated phytohormone or controls. We collected root-influenced soil adjacent to plant roots (<20 cm) from two different cropping systems: common bean (*Phasoelus vulgaris*) and switchgrass (*Panicum virgatum*). These two plants were chosen for their distinct life strategies (annual and perennial) and families (Fabaceae and Poaceae), thereby providing a contrast for understanding the potential generalizability of microbiome responses to phytohormones. We used field soils to retain the legacy of the crop in assembling the bacterial communities relevant to agricultural conditions. We applied the stress-associated phytohormones abscisic acid (drought) and salicylic acid (pathogen infection) to the mesocosm soils. We used 16S rRNA and rRNA gene amplicon sequencing to designate a taxon-level activity state (likely active, likely inactive) before and after phytohormone application. We also related microbiome data to soil nutrient concentrations to understand their impact on shifts in microbial community activity across treatments and over time.

## MATERIALS AND METHODS

### Soil collection, processing, and storage

Five kilograms of root-zone soil (within 10–15 cm from roots) was collected from four replicated switchgrass plots at Kellogg Biological Station's Great Lakes Bioenergy

Research Center Bioenergy Cropping Systems Experiment in November 2019. Approximately 4 kg of soil was also collected from four replicate rotation soil plots growing common beans (*Phaseolus vulgaris* L.) at Michigan State University's Agronomy Farm in East Lansing, in November 2019. Approximately 1,000 g of soil from each plot was collected from the top 10 cm using an ethanol-sterilized soil auger with a 10 cm diameter and placed in Whirlpak bags. Similar to the switchgrass soil sampling, soils were sampled within 10–15 cm of the roots of the bean crops to maximize the root influence in the collected soils. Soils were transported to the lab in a cooler with ice and then stored at 4°C until further processing.

The following day after sample collection, soils from replicate plots were composited into ethanol-sterilized buckets and hand-mixed to homogenize. Composited soils were sieved through a 4 mm mesh to remove roots and debris. Three 0.5 g subsamples of each composited soil were flash-frozen in liquid nitrogen for active microbiome assessment and designated as the "field-soil" samples. The remaining sieved soil was stored at 4°C until the start of the experiment. All flash-frozen soil samples were stored at −80°C.

The soil organic matter, pH, lime index, phosphorus, potassium, calcium, magnesium, nitrate, and ammonium levels of the "field-soil" soils were analyzed by the Michigan State Soil, Plant, and Nutrient Laboratory according to their standard protocols (Table 1). All soil chemical tests followed the methods described in (41), except for those quantifying nitrate-nitrogen and ammonium-nitrogen concentrations, which were determined using the LaChat QuikChem 8500 Flow Injection Analyzer (Lachat Instruments, Loveland, CO) as described in reference 42 for nitrate-N and reference 43) for ammonium-N. We briefly note the procedure for each chemical analysis as follows. Lime requirement determination was conducted using the Shoemaker-McLean-Pratt (SMP) buffer method, and pH was analyzed using an AS3000 pH Analyzer (Labfit, Bayswater, Australia). Phosphorus concentration was determined using the Bray P1 extracting solution for soil extractions; acid molybdate, ascorbic acid, and a molybdate working solution for dilution and color development. Phosphorus concentration was determined by preparing a stock standard phosphorus solution (1000 ppm) and generating a phosphorus standard curve. Measurements were conducted using the Brinkmann PC950 Probe Colorimeter (Enapart Industrial Supplies, San Antonio, TX). Potassium and calcium concentrations were determined by flame emission, while magnesium concentration was measured colorimetrically using an autoanalyzer (AutoAnalyzer3, SEAL Analytical, Milwaukee, WI). The percentage of soil organic matter was determined using the loss-on-ignition method.

Gravimetric soil moisture was determined by comparing the mass difference between the initial field soil sample and a sample dried at 92°C for 4 days. Water holding capacity (WHC) was assessed by taking 10 g of dry soil and placing it in a funnel lined with filter paper (44). The dry soil was saturated with water and allowed to drain for 2 h. The mass of the original soil, the mass of water added, and the mass of water drained were used to calculate WHC as percent gravimetric soil moisture. Before setting up the mesocosms, 0.5 g of stored soil was flash-frozen, denoting a "pre-dry" condition that accounted for any shifts in the microbial community due to storage, thereby capturing a true baseline reference community. The first step of the experiment was drying the soils to induce dormancy and use them as a reference for subsequent treatments. Soils were dried in a 55°C incubator for 3 days, and after drying, soil subsamples were flash-frozen as the "post-dry" treatment before adding water and phytohormones.

**TABLE 1** Nutrient analysis of bean and switchgrass root zone soil before the experiment (pre-dry sample)[a]

| Soil | pH | Lime Index | Bray P1-P ppm | K ppm | Ca ppm | Mg ppm | OM % | $NO_3$-N ppm | $NH_4$-N ppm |
|---|---|---|---|---|---|---|---|---|---|
| Switchgrass | 6.0 | 69 | 27 | 96 | 930 | 146 | 2.5 | 1.5 | 2.6 |
| Common bean | 6.2 | 69 | 97 | 151 | 486 | 82 | 1.9 | 0.7 | 1.6 |

[a]P-Phosphorus, K-Potassium, Ca- Calcium, Mg-Magnesium, OM-organic matter, $NO_3$-N:Nitrate-N, $NH_4$-N: Ammonium-N. Bray P1-P refers to Phosphorus content (ppm) measured using the Bray Kurtz P1 (weak acid) test.

## Experimental design

There were four experimental conditions, including controls: methanol carrier control (water and methanol), abscisic acid (water, methanol, and abscisic acid), salicylic acid (water, methanol, and salicylic acid), and a water control (water only), and three time points (Day 1, Day 7, and Day 14). Each condition was replicated four times for 16 mesocosms (four conditions, four replicates) for each soil type (Fig. 1). This design was executed for two soils (bean and switchgrass root zone), resulting in 32 mesocosms. Including the field soil, pre- and post-dry samples, water acclimation, and the time series for each experimental condition, there were 333 microbiome samples across the bean and switchgrass soils.

## Mesocosm set-up, phytohormone treatment, and maintenance

To create each mesocosm, 100 g dried soil was adjusted to 25% WHC and placed into 473 mL sterile glass jars (Fig. 1). Mesocosms were maintained at 27°C with lids secured loosely to ensure aerobic conditions. Mesocosm manipulations and samplings were performed inside a sterilized biosafety cabinet. Every 2–3 days, mesocosms were massed, and any water loss was replaced with sterile water to maintain soil moisture.

Before hormone addition, mesocosms were incubated for 2 weeks to acclimatize to the baseline water content of 25% WHC. This water acclimatization step was included to ensure that changes in activation could be attributed to the experimental treatment of hormone addition and not exclusively to water addition from a dry state, as it has been well-reported that soil microorganisms activate in response to water (45–48) After 2 weeks, three 0.5 g subsamples from four replicate mesocosms designated as "post 25% WHC" were flash frozen in liquid nitrogen and designated as "water acclimated" samples. After sample collection at 2 weeks, the "post 25%" mesocosms were supplemented with the required water to reach 50% WHC. They were designated as "water control" samples, while the rest of the mesocosms received phytohormones dissolved in methanol and methanol alone (carrier control) mixed with enough water to reach 50% WHC.

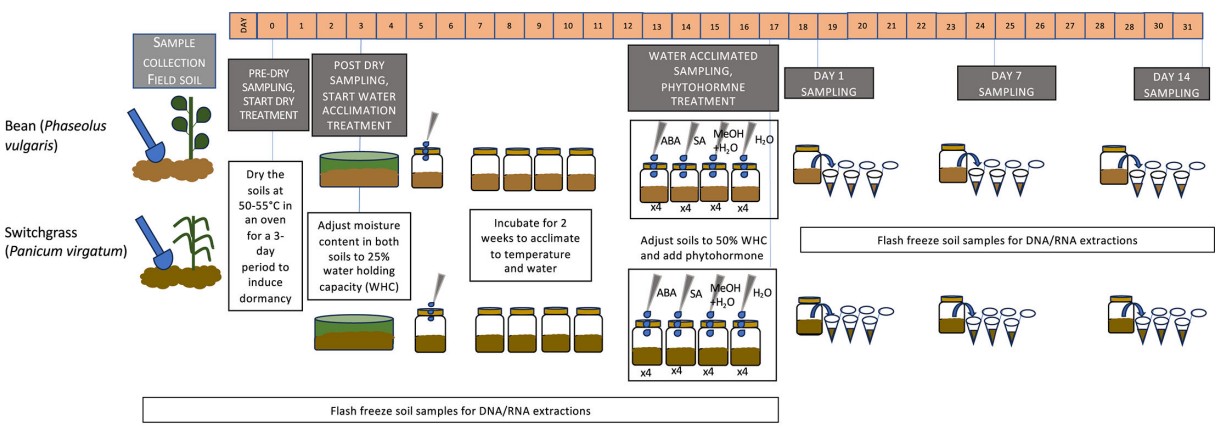

Full factorial design with 32 experimental units = 2 soil types * 4 treatments * 4 replicates (Treatments: ABA, SA, control with methanol + water, control only water)

**FIG 1** Schematic of the soil mesocosm experimental design and sampling approach to assess how root zone bacterial communities activate and assemble in response to phytohormones that indicate plant stress. The design was a time series of field-collected root zone soils that were first deactivated by drying and then acclimated to water before the experimental addition of phytohormones. The water acclimation step was used to partition the response to moisture increase from the response to the phytohormones. Soils were amended with either a stress-associated phytohormone (salicylic acid, SA, or abscisic acid, ABA) or a methanol carrier or water control. WHC is water holding capacity. All soil samples collected (field soil to end of 14-day experiment) were flash frozen in liquid nitrogen for DNA/RNA extractions. Sample size used in DNA/RNA analyses at each time point: field soil bean (*N* = 2), field soil switchgrass (*N* = 3), pre-dry bean (*N* = 2), pre-dry switchgrass (*N* = 2), post-dry bean (*N* = 5), post-dry switchgrass (*N* = 3); after water acclimation treatment (2 weeks stabilization), 12 total samples were collected each from bean and switchgrass mesocosms (4 mesocosm replicates times 3 technical replicates = 12 samples), totaling 12 samples *2 soil types = 24 samples. Similarly, 12 samples were collected for each soil/treatment combination (bean ABA, bean SA, switchgrass ABA, switchgrass SA) at three time points (Day 1, Day 7, and Day 14) totaling 12*4*3 = 144 samples.

Phytohormones were dissolved in methanol to create a 0.3 M solution, and 1 mL of the phytohormone-methanol solution was filter sterilized (0.2 µm) before adding to the mesocosms in combination with enough water to achieve 50% WHC. We normalized the hormone addition based on the amount of carbon added per g of dry soil, and a 3 M solution for ABA and SA resulted in approximately 250–500 µg C g-1 dry soil for SA and ABA. Phytohormone concentrations reported in soil are typically in the nanomolar range (49). However, we chose a concentration that was 4–7 times the reported values used in the literature (which range from 5 to 64 µg C g-1 dry soil [30, 50]), to elicit a measurable response to the phytohormone, such that comparisons across treatments would be possible. Methanol carrier controls received 1 mL of methanol supplemented with water to achieve a 50% WHC. After adding the aqueous solution, the mesocosm soil was mixed using a sterile spatula. The soil was non-destructively sampled from each mesocosm and immediately flash-frozen on days 1, 7, and 14 after hormone addition.

## RNA/DNA coextractions

A manual phenol-chloroform nucleic acid extraction was performed to obtain RNA and DNA from the same cell lysis pool (51) with minor modifications. For the modifications, we used Qiagen Powerbead Garnet Tubes (0.70 mm). After a 30 s lysis in a bead beater, we performed a 10 min centrifugation at 12,000 $g$ at 4℃. After adding chloroform-iso-amyl alcohol (24:1), the tubes were inverted several times to form an emulsion, followed by 5 min of centrifugation at 16,000 $g$ and 4℃. After adding 30% PEG6000-1.6 M NaCl, the tubes were inverted several times to mix the contents and then placed on ice for a 2 h incubation. After incubation, the samples were centrifuged for 20 min at 16,000 $g$ at 4℃. After centrifugation, the supernatant was removed, and 1.0 mL of 70% EtOH (stored at −20℃ overnight) was added. The samples were centrifuged for 20 min at 16,000 $g$ at 4℃, and the ethanol wash was removed. The samples were centrifuged for an additional 20 s. Residual ethanol was removed with a pipette, and the pellet was left to air-dry before they were resuspended in 30 µL of nuclease-free water. Negative extraction controls (tubes with reagents and beads) were processed alongside experimental samples. Nucleic acids were visualized using agarose gel electrophoresis and validated by the presence of a distinct band for DNA and RNA (51). All DNA and RNA samples were quantified using a Qubit dsDNA BR assay kit and RNA HS assay kit on a Qubit 2.0 fluorometer (Invitrogen, Carlsbad, CA, USA). Samples were stored at −80℃.

## DNase treatment and cDNA synthesis

RNA samples were purified using an Invitrogen TURBO DNA-free Kit with a slightly modified protocol. A six microliters aliquot of the nucleic acid coextraction was treated with 1 µL of 10× TURBO DNase Buffer and 3 µL of Turbo DNase enzyme and then incubated at 37℃ for 30 min. After this incubation period, 2 µL of DNA inactivation reagent was added to each tube and the mixture was incubated at room temperature for 5 min. Then, the sample was centrifuged at 2,000 $g$ for 5 min at room temperature. The purified RNA was used as a template to form cDNA using Invitrogen's SuperScript III First-Strand Synthesis System using random hexamers per kit directions and with negative controls to assess for reagent contamination. Final cDNA samples were stored at −80℃.

## PCR check and gel electrophoresis

16S rRNA gene V4 PCR was performed for every DNA and cDNA sample to ensure the presence of the 16S rRNA gene and the RNA sample to ensure effective DNase treatment. The PCR used a 2× GoTaq Green Master Mix and the primers 515F (GTGY-CAGCMGCCGCGGTAA) and 806R (GGACTACNVGGGTWTCTAAT). PCR cycle parameters were 95℃ for 3 min; 30 cycles of 95℃ for 45 s, 50℃ for 60 s, 72℃ for 90 s, and a final step at 72℃ for 10 min. The PCR product was examined via gel electrophoresis. All PCRs included a no-template negative control and an *E. coli* DNA template as the positive control.

## Illumina sequencing of the 16S rRNA gene and 16S rRNA

DNA and cDNA samples were sent for 16S rRNA gene V4 amplicon sequencing at the Genomics Research Technology Support Facility at Michigan State University. The V4 hypervariable region of the 16S rRNA gene was amplified using dual-indexed Illumina-compatible primers 515f/806r, as described by Kozich et al. (52). PCR products were batch normalized using an Invitrogen SequalPrep DNA Normalization Plate, and the normalized products recovered were pooled. The pool was cleaned and concentrated using a QIAquick PCR Purification column, followed by AMPureXP magnetic beads. It was then quality-controlled and quantified using a combination of Qubit dsDNA HS, Agilent 4200 TapeStation HS DNA1000, and Invitrogen Collibri Library Quantification qPCR assays. The pool was loaded onto an Illumina MiSeq v2 standard flow cell, and sequencing was performed in a $2 \times 250$ bp paired-end format using a MiSeq v2 500-cycle reagent cartridge. Base calling was performed using Illumina Real-Time Analysis (RTA) v1.18.54, and the RTA output was demultiplexed and converted to FASTQ format with Illumina Bcl2fastq v2.20.0.

Sequence data were analyzed using QIIME2 (53). Briefly, all paired-end sequences with quality scores were compressed and denoised using the DADA2 plugin (54). The denoising step dereplicated sequences, filtered chimeras, and merged paired-end reads. The truncation parameters for DADA2 were determined using FIGARO (55) developed by Zymo Research Corporation. All truncation was performed from the 3′ end for consistent final read lengths. The DNA and cDNA data sets were separately quality controlled and denoised because the additional cDNA amplification step could have introduced different errors for the cDNA than for the DNA amplicons. The quality-controlled DNA and cDNA count tables were merged into a single QIIME2 artifact using the feature-table merge command. Similarly, the DNA and cDNA representative sequences were merged into a single QIIME2 artifact using the feature-table merge-seqs command. The representative sequences from the combined count tables were clustered at 99% identity using a *de novo* approach, and the clustered representative sequences were then classified using SILVA v138 (56) to generate the taxonomy file. Again, 99% sequence identity was used to conservatively account for possible errors in the cDNA amplicons resulting from the additional amplification (RT-PCR). The resulting OTU table and taxonomy files were exported to R for ecological analysis.

## Designating the active bacterial populations

All downstream analyses were performed in R version 4.0. The R package decontam (57) was used to determine the number and identity of contaminants in the data set and remove them using the prevalence method. Contaminating taxa, mitochondria, and chloroplast sequences were filtered from the data sets. A subsampling depth of 12,000 reads per sample was selected based on rarefaction curves. After subsampling, 16S rRNA to rRNA gene ratios were computed from the cDNA and DNA data sets described in Bowsher et al. (38). We chose the method that applied a 16S rRNA:rRNA gene ratio threshold $>= 1$. We also included "phantom taxa" and changed all DNA counts $= 0$ corresponding to RNA counts $> 0$ to DNA counts $= 1$. The DNA counts table was then filtered to include only taxa meeting the criteria to be likely active at the population average (e.g., 16S rRNA:rRNA $>= 1$). We used DNA counts instead of RNA counts of likely active taxa to limit biases in relative transcription levels across taxa in the community. The filtered DNA OTU table was used for all ecological analyzes, and we refer to this as the "active community."

## Ecological statistics

Microbiome analysis was performed in R using the phyloseq (58) and vegan (59). Permutational analysis of variance (PERMANOVA) was used for multivariate statistics. For data visualization, we used a constrained analysis of principal coordinates (CAP) and partitioned out the variances contributed by selected variables. Specifically, switchgrass

had a significant mesocosm effect (PERMANOVA pseudo-$F$ = 2.84, $R^2$ = 0.05, $P$ = 0.002), so to account for that, we partitioned out the mesocosm variance using a constrained ordination. We also used soil chemistry data to conduct a constrained analysis of principal coordinates to understand how soil nutrient status affected microbiome community changes across treatments using the package phyloseq. Edaphic factors included in the final models were selected with function ordistep from the vegan package, with the initial full model incorporating pH, percent soil organic matter, nitrate, and ammonium concentrations. We used differential expression analysis (DESeq2 [60]) to assess the log twofold changes between time points across treatments. The Benjamini-Hochberg procedure (61) was used to correct raw $P$-values, adjusting for the false discovery rate (FDR). An FDR cut-off of 0.05 was used to identify significant changes between time points. We looked at each taxon's relative abundance separately to determine the reactivation of taxa from an inactive state. Activity dynamics were evaluated using the methods described in Bandopadhyay et al. (40). In this study, we did not set a threshold for detecting phantom taxa across samples. To generate the heatmaps and bubble plots looking at relative abundances for each taxon, we used a max standardization approach using the decostand function in the vegan package in R (59). For alpha diversity, we used the estimate_richness function in the package phyloseq with ANOVA type III results reported using the R package car (62). To assess if individual bacterial classes significantly differed by treatment, we used a Kruskal-Wallis rank sum test.

## RESULTS

### Sequencing summary

We sequenced a total of 333 DNA samples and 333 cDNA samples from this experiment. After rarefying all read libraries to 12,000 reads per sample (Fig. S1A), 324 DNA and 324 cDNA samples met the minimum observation effort. After removing all contaminants from the OTU table using the package decontam (Fig. S1B and C), 13,138 OTUs were used to determine the active community.

### Microbial communities differed across the phytohormone treatments and timepoints, with decreased alpha diversity observed for specific phytohormone treatments

There was a generally consistent response of the active bacterial microbiome's structure (beta diversity) to phytohormone addition, regardless of the soil origin (Fig. 2; Table 2). Phytohormone treatment (PERMANOVA bean, pseudo-$F$ = 10.28, $P$ = 0.001, switchgrass pseudo-$F$ = 9.24, $P$ = 0.001) and time (PERMANOVA bean, pseudo-$F$ = 6.63, $P$ = 0.001, switchgrass pseudo-$F$ = 12.79, $P$ = 0.001) both explained most global variation in the active community structure. When excluding the three baseline samples (field, pre-dry, post-dry), we found that bacterial community richness was statistically different between crop soil (ANOVA $F$ = 36.73, SumSq = 46,434, $P$ < 0.001) and time point (water acclimated, Day 1, Day 7, and Day 14) (ANOVA $F$ = 8.22, SumSq = 31,165, $P$ < 0.001). In contrast, Inverse Simpson's diversity, which accounts for richness and evenness, differed by time point (ANOVA $F$ = 3.04, SumSq = 35,840, $P$ = 0.03) but not by crop soil. There was also a strong interaction effect between crop soil and treatment (abscisic acid, salicylic acid, methanol control, and water control) on bacterial community richness (ANOVA $F$ = 12.78, SumSq = 32,290, $P$ < 0.001). At the same time, treatment strongly influenced Inverse Simpson's diversity (ANOVA $F$ = 4.81, SumSq = 47,335, $P$ = 0.003). Overall, we found that richness and diversity decreased over time for switchgrass soils treated with abscisic acid (Fig. S2).

Bean and switchgrass soils differed distinctly in their active community structure across all time points and treatments (PERMANOVA $F$ = 29.05, $R^2$ = 0.08, $P$ = 0.001). For the ABA treatment, there was a higher proportion of Bacilli (Kruskal-Wallis chi-squared = 9.36, $P$ = 0.002) and Verrucomicrobiae (Kruskal-Wallis chi-squared = 4.71, $P$ = 0.03)

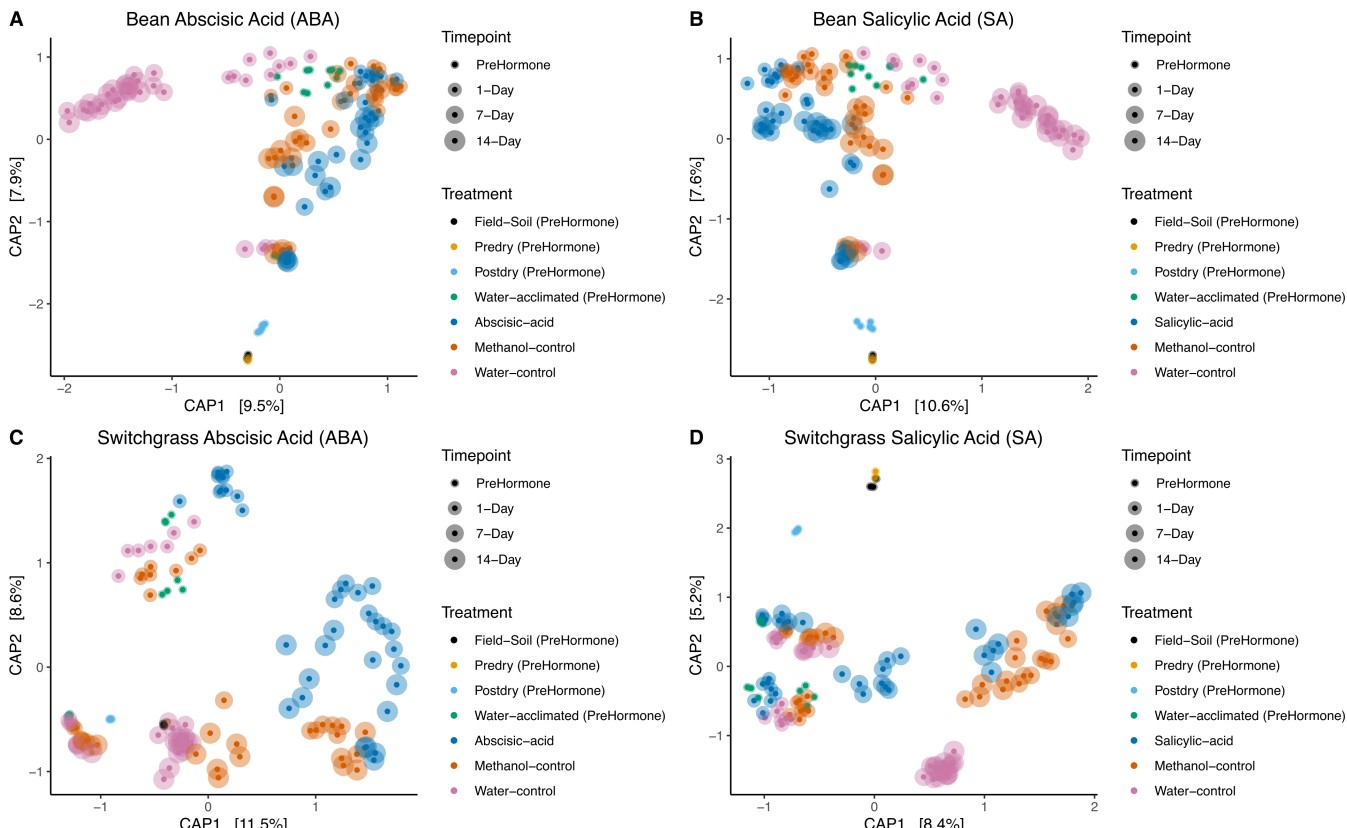

**FIG 2** Active microbiome community structure differs across time points for (A) bean soil amended with abscisic acid (ABA), (B) bean soil amended with salicylic acid (SA), (C) switchgrass soil amended with abscisic acid (ABA), and (D) switchgrass soil amended with salicylic acid (SA) treatments. Ordination shows a constrained analysis of principal coordinates (CAP) after excluding the variance attributed to mesocosm. The increasing symbol sizes denote sampling time and the colors denote the treatments.

in bean soils on Day 1 after ABA addition, but a higher proportion of Actinobacteria (Kruskal-Wallis chi-squared = 8.33, $P = 0.004$) and Myxococcia (Kruskal-Wallis chi-squared = 17.28, $P < 0.001$) in switchgrass soils. Then, Alphaproteobacteria increased proportion 7 (Kruskal-Wallis chi-squared = 7.05, $P = 0.008$) and 14 days (Kruskal-Wallis chi-squared = 5.19, $P = 0.02$) after ABA exposure in switchgrass compared to bean soils (Fig. 3). Actinobacteria generally declined in switchgrass soils over time and were generally replaced by Alphaproteobacteria. In bean soils, active Actinobacteria increased over time, with a concurrent relative decline in several other bacterial classes (Fig. 3).

We also related changes in the active bacterial communities at 14-day to measured soil properties at 14-day time point (Table S1) to consider any co-occurring factors (Fig. S3). We found that any consistent phytohormone responses by microbial taxa were unlikely to be attributable to concurrent pH or organic matter changes because the treatments generally had different properties (Fig. S5). Nitrate (in bean) and ammonium (in switchgrass) explained the variation in the soil communities after receiving water (Fig. S5). Overall, soil differences in pH, nitrate concentrations, and organic matter content

**TABLE 2** Permuted analysis of variance (PERMANOVA) to assess global differences in active microbial community structure across experimental treatments and time points

| | Bean | | | | Switchgrass | | | |
|---|---|---|---|---|---|---|---|---|
| | Degrees of freedom | $R^2$ | Pseudo-$F$ value | $P$ value | Degrees of freedom | $R^2$ | Pseudo-$F$ value | $P$ value |
| Treatment | 4 | 0.19 | 10.28 | 0.001 | 4 | 0.16 | 9.24 | 0.001 |
| Time point | 2 | 0.06 | 6.63 | 0.001 | 2 | 0.11 | 12.79 | 0.001 |
| Treatment: Time point | 6 | 0.10 | 3.66 | 0.001 | 6 | 0.11 | 3.98 | 0.001 |

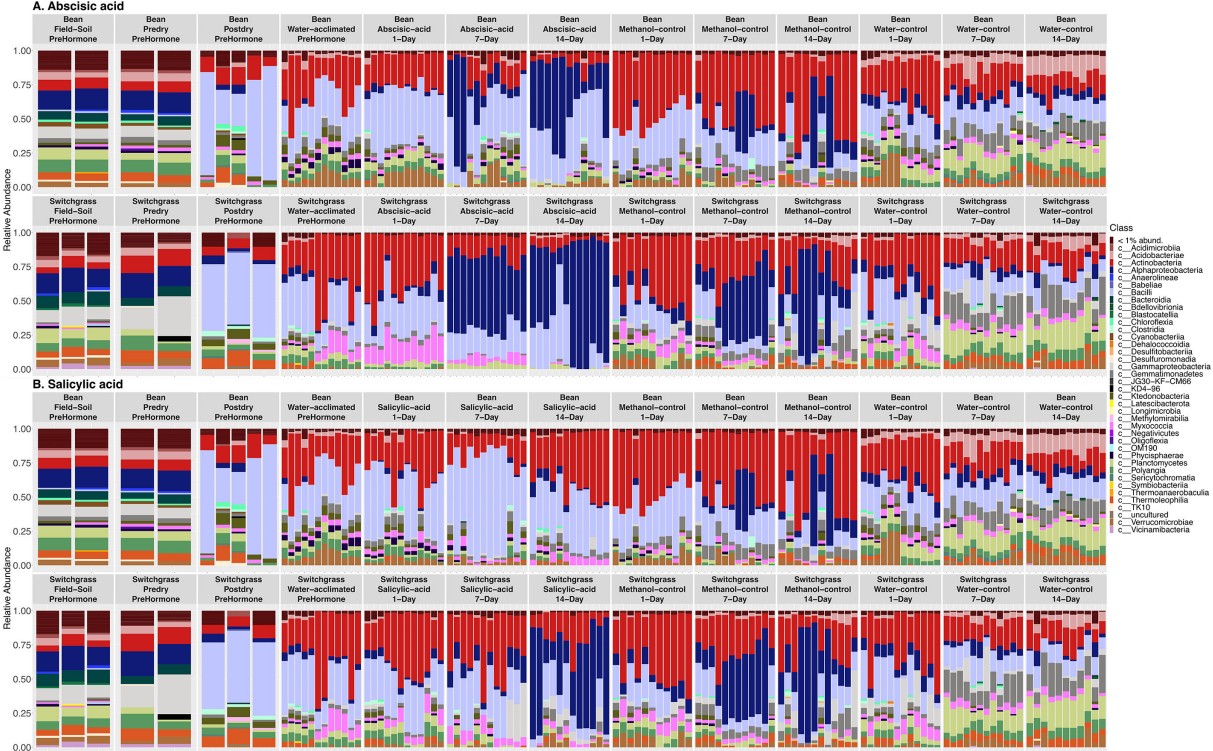

**FIG 3** Active bacterial community structures differed between bean and switchgrass. Barplots show broad changes in the bacterial community composition over time and across treatments in bean and switchgrass root zone soils for (A) abscisic acid and (B) salicylic acid.

explained about 27% of the variation in the active bacterial community in bean soils. In contrast, pH, ammonium concentration, and organic matter content explained about 22% of the variation in the active bacterial community in switchgrass.

## Immediate activation and persistence of the root zone bacterial microbiome to phytohormones

After determining the active taxa at Day 1 post-hormone addition, we assessed their relative abundances (16S rRNA gene read counts) before and after phytohormone addition. To focus on enrichments observed uniquely in the phytohormone treatments, we also compared the enrichment of specific taxa after hormone addition (Day 1) to the water-acclimated sample and the water and methanol controls. There were statistical differences in the active bacterial community structure between the water-acclimated and Day 1 for bean ABA soil (PERMANOVA $F = 3.95$, $R^2 = 0.15$, $P = 0.002$), bean SA soil (PERMANOVA $F = 3.96$, $R^2 = 0.15$, $P = 0.006$), switchgrass ABA soil (PERMANOVA $F = 10.50$, $R^2 = 0.32$, $P = 0.001$), and switchgrass SA soil (PERMANOVA $F = 2.31$, $R^2 = 0.09$, $P = 0.002$).

In the ABA-treated soils, three OTUs were enriched by a log twofold increase at Day 1 compared to the water-acclimated sample, but not in the water and methanol controls (Fig. S4A). These included one OTU from Family Myxococcaceae (log twofold enrichment = 1.57, p-adj = 0.01), and two actinobacterial OTUs belonging to genus *Microbispora* (log twofold enrichment = 5.1, p-adj = 0.003), and genus *Streptomyces* (log twofold enrichment = 4.9, p-adj < 0.001). However, these enrichments were detected only in the switchgrass soils, and there were no OTUs enriched in bean soils treated with ABA on Day 1. Furthermore, all three taxa enriched in the switchgrass ABA soils switched from either an inactive or below detection state in the water-acclimated sample to high abundance at Day one after hormone addition (Fig. S5B).

Similarly, for the SA treatments, one OTU belonging to genus *Microbispora* was log twofold enriched at Day 1 (log twofold enrichment in bean = 6.78, p-adj = 0.0005,

switchgrass = 5.44, p-adj = 0.005) compared to the water acclimated sample in both bean and switchgrass and was enriched solely in the SA treatments to the exclusion of the methanol and water controls (Fig. S4B). While this OTU was enriched on Day one in both bean and switchgrass (Fig. S5C and D), another OTU characterized as *Microbispora* was enriched on Day 1 in SA- treated bean soils only (log twofold enrichment = 7.1, p-adj = 0.0003), Furthermore, the response pattern suggested a likely switch from inactive (detected in DNA but not RNA) to active for both taxa.

Several taxa belonging to classes Myxococcia, Actinobacteria, Alphaproteobacteria, and Bacilli were resuscitated within 1 day after SA addition to the bean soil (Fig. S5C), but most of these taxa were also responsive to methanol (Fig. S5E). Two actinobacterial OTUs (both belonging to the genus *Microbispora*) were significantly activated in response to SA treatment in bean soils and not in the methanol controls. As mentioned above, these two OTUs were log two-fold enriched after Day 1 of SA exposure compared to the water-acclimated soils. However, only one of these *Microbispora* OTUs was represented in the top 50 abundant and active taxa for the bean SA treatment, but both are included

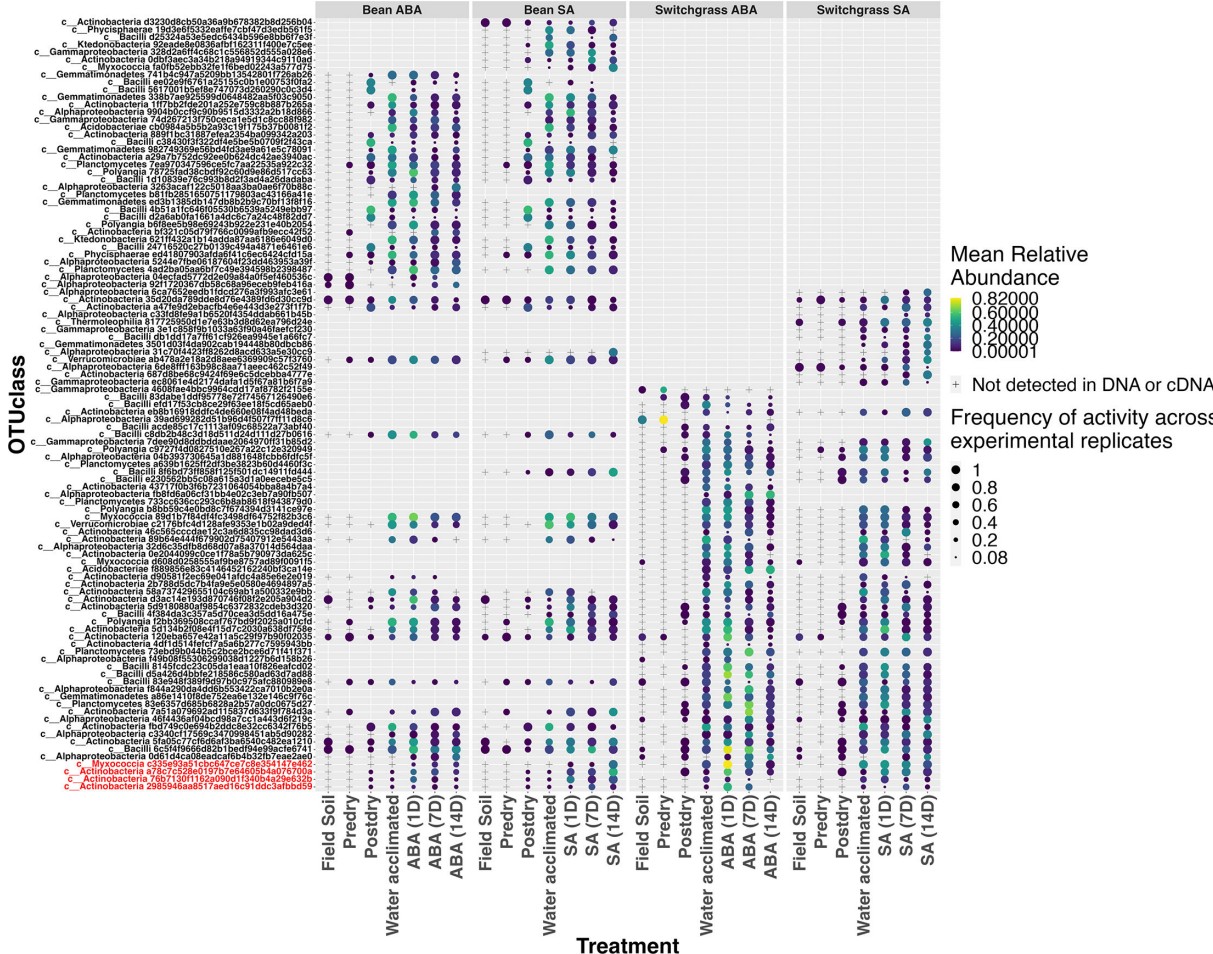

**FIG 4** Bacterial microbiome community activation, persistence, and assembly before and after phytohormone addition to bean and switchgrass root zone soil. The relative abundances of active taxa that were log twofold enriched (DESeq analysis) at Day 1 as compared to the water-acclimated sample are shown in red. All other taxa include the 50 most abundant active members in each soil/phytohormone condition. All responses shown were specific to a phytohormone exposure (salicylic acid, SA, or abscisic acid, ABA) and were not observed in controls. Inactive taxa have a relative abundance of zero, so they do not contain a symbol on the graph and are left blank for that time point/treatment. Taxa not detected in either DNA or cDNA data set are denoted by a + shape. Color gradient denotes increased relative abundances as color changes from blue to yellow. The shape size of each bubble denotes the consistency of detection across experimental and technical replicates of soil mesocosms (e.g., with 1 denoting activity in all replicates and 0.5 denoting activity in half). Empty rows indicate that the taxon was not detected within the top 50 most abundant members for that condition. Empty spaces in rows that also have symbols indicate that the taxon was not classified as active within those samples (columns) but was detected in DNA.

for comparison (Fig. S5C). Both taxa were detected as active in methanol controls but not as consistently enriched as in the SA treatment. Among the most abundant active members in bean soils treated with ABA and SA, we identified an Alphaproteobacterial OTU (*Microvirga*) that activated within 1 day after hormone addition but did not activate in the methanol control mesocosms. This taxon was not captured in the DeSeq results, likely due to a non-significant increase in abundance post-hormone exposure. However, apart from the log twofold enrichments mentioned above, no substantial response to ABA or SA was noted among other active OTUs for switchgrass beyond their similar response to the methanol on Day 1.

While several active taxa increased abundances after phytohormone addition (Fig. 4), many of these were also active in the methanol controls (Fig. S6). However, some active taxa persisted past Day 1, a pattern consistent with growth if the phytohormones were used as a resource. We observed the persistence of active taxa over 14 days for four of the enriched taxa (indicated in red in Fig. 4), including a *Microbispora* for switchgrass and bean SA and switchgrass ABA, and a Myxococcaceae and *Streptomyces* in switchgrass ABA. Notably, persistent activation over time could also be an indirect response to the phytohormone addition (e.g., due to priming), which would not be distinguishable here.

> ***Revisiting hypotheses:*** *The microbiomes of root-zone soils from different plants have different activation and assembly dynamics in response to phytohormones, though some taxa show generic trends across crops and phytohormones*

Several taxa were enriched on Day 1 after ABA addition in switchgrass soils compared to the water-acclimated control. Specifically, OTUs belonging to phyla Actinobacteriota (a *Streptomyces* sp.) and Myxococcota were enriched by log twofold on Day 1 compared to the pre-hormone time point (Fig. S4). Bean soil, however, showed no taxa enriched on Day one in response to ABA. We also found evidence of a *Microbispora* OTU that was specifically enriched in bean soil in response to SA treatment (Fig. S4). Thus, we found that soils carrying the legacy of growing a specific crop exhibited distinct responses in microbial activation to specific phytohormones, supporting Hypothesis 1.

However, one *Microbispora* OTU was enriched in response to both ABA and SA in switchgrass soils on Day 1 compared to the water-acclimated sample. This same OTU was also enriched in bean soils in response to SA. Thus, this lineage responded consistently across different soils and phytohormones. This suggests that certain bacterial lineages can generally activate in response to phytohormones regardless of the crop legacy, supporting hypothesis 2. This also suggests that members of the same genus (e.g., *Microbispora*) exhibit varying trends across phytohormone treatments and crops, with some showing specificity to a particular phytohormone and crop. In contrast, others display generic trends across phytohormone types and crops.

Although we identified hormone-specific and crop-specific responses in specific taxa, we found no strong evidence of any taxa common to both plants' root zone soils that were similarly responsive to either ABA or SA exposure, and thus, H3 was not supported.

## DISCUSSION

This study assessed the activation and assembly dynamics of activated root zone bacterial taxa in response to phytohormones produced by plants during stress. Thus, this study provides new insights into which plant-associated bacterial populations immediately activate in response to SA and ABA phytohormones, suggesting that these molecules could serve as reactivation signals. Furthermore, this study also distinguished general and specific activation dynamics of bacterial taxa for two different plants' root zone soils over two weeks after exposure, allowing improved precision on the ecological underpinnings of these responses. Ultimately, we can weigh the evidence that the activated taxa responded to the phytohormones as a signal to a plant's "cry for help."

In this study, crop-specific responses of the root zone microbiome to phytohormones indicated that the plant legacy in its associated soil partially determines the microbiome's response, providing evidence in support of H1. Given that most plant species harbor

unique microbial communities (63–65), it is not unreasonable that each plant species may have a different microbiome response to stress indicators like phytohormones. While switchgrass responded to ABA via immediate enrichment of three OTUs within 24 h of phytohormone application, bean soils showed no enrichment of any taxa in response to ABA. Similarly, one actinobacterial OTU was enriched in response to SA treatment only in bean but not in switchgrass soils, suggesting a legacy effect that impacts the recruitment of active members to the bean rhizobiome. This finding aligns with the results observed in our previous study, which assessed bacterial reactivation after drought (40). Other studies have also shown that root exudate-induced soil respiration can be observed nearly immediately within 6–21 h (30). However, we note that the crop-specific trends observed may be correlated with other soil properties, such as soil type, texture, location, and management practices, as the field sites used to collect root-zone soils of switchgrass and common bean differed. Therefore, it is not possible to determine any crop-specific effects or soil-specific effects here.

Actinobacterial genera are widely reported to be associated with drought stress in plants (66), so it was not surprising that we detected several actinobacterial taxa that activated in response to ABA and SA exposure, showing support for H2 among these lineages. Among the most abundant active members in bean soils exposed to ABA and SA, we identified an Alphaproteobacterial OTU (*Microvirga*) that activated within 24 h after hormone addition, but did not activate in the methanol control mesocosms within the same time frame. The Alphaproteobacterial genus *Microvirga* has also been shown to alleviate drought stress in plants, such as cowpeas. Cowpeas inoculated with *M. vignae* exhibited no differences in stomatal conductance, shoot dry mass, or N accumulation between water deficit and well-watered conditions (67). This same study found that ABA biosynthesis and ABA-dependent genes were less upregulated under drought when plants were inoculated with a *Microvirga sp.* compared to a *Bradyrhizobium sp*, suggesting that the presence of *Microvirga* relatively decreased the plant's typical ABA production during stress.

We identified a specific actinobacterial genus, *Microbispora*, that consistently responded to both ABA (switchgrass) and SA (in bean and switchgrass). This response pattern indicates that this microbial group could be a generic responder to phytohormones as stress signals. Actinobacteria have been widely explored as targets for developing bioinoculants. *Microbispora* and *Streptomyces* have been suggested to be both soil-inhabiting and endophytic genera though only a limited number of species are likely to have this dual habitat preference (68). *Microbispora* and other actinobacterial species, including those belonging to *Actinoplanes, Streptomyces, Rhodococcus*, and *Micromonospora,* have also been explored for their direct/indirect plant growth-promoting capabilities, which involve the secretion of phytohormones, siderophores, and antimicrobial agents (69, 70).

Across the two soils used in this study (bean and switchgrass), we found no taxa that consistently responded to either ABA in bean and switchgrass or SA in bean and switchgrass, indicating that hormone-specific responses may not be conserved across different plants, and thus H3 was not supported. We would like to note that the *Microbispora* sp. enriched in SA-treated bean and switchgrass soils was also detected in ABA-treated switchgrass soils; therefore, we do not consider this a SA-specific response. Besides this specific *Microbispora* sp., the OTUs enriched in ABA-treated switchgrass soil or SA-treated bean soils were not enriched in the bean and switchgrass soils, respectively. However, we did find hormone-specific responses to either ABA or SA within each soil type. For example, we found one actinobacterial species (*Microbispora*) that uniquely responded to SA in bean, while one actinobacterial species (*Streptomyces*) and one myxoccocal OTU responded uniquely to ABA in switchgrass. While we found signatures of Gram-positive bacteria activating to phytohormones, other studies have found a declining abundance of Gram-positive biomarkers following phytohormone addition (29). However, these other studies did not discriminate between activity states but rather assessed microbial communities using phospholipid-derived fatty acid profiles. A

negative response to phytohormones may also be due to an investment in complementary survival strategies, such as osmolyte production and spore production, rather than growth and turnover (71–73).

The literature shows that phytohormone signaling can play a major role in rhizobiome assembly (25). For instance, using natural selection, Kalachova et al. (74) showed a reduction in disease severity of Arabidopsis plants infected with foliar pathogen *Pseudomonas syringae*. This was associated with SA-mediated induction of defense mechanisms and a shift in the soil bacterial community, but not in the fungal community. While one study showed that ABA and polyacrylamide treatments to soil coupled with foliar application of ABA reduced the relative abundances of Actinobacteria in the rhizosphere using amplicon DNA sequencing and increased drought resistance in forage grass (75), other studies have shown that Actinobacteria increase in plant roots and rhizosphere soil during drought likely due to expansion of C allocation in root exudates (66).

Furthermore, certain Actinobacteria, such as *Streptomyces,* have been shown to increase the ABA content of wheat leaves to upregulate the expression of drought-resistance genes (76). Thus, there could be implications for a positive feedback loop in actinobacterial responses to a plant's cry for help. The role of ABA as a carbon substrate for Actinobacteria is also indicated. For example, *Rhodococcus* sp. utilized ABA as the sole carbon source in batch culture (77). These findings support the idea that phytohormone substrates can be used to select rhizosphere microbes (78) and corroborate our finding of active Actinobacteria in this study. Overall, there are limited experiments that consider the reactivation of the microbiome through plant signaling.

One of the major challenges in the development of bioinoculants is the continued persistence of beneficial microbes in the soil during and after stress events, such as drought and phosphorus limitation (79–81). Through this study, we differentiate between certain stress-responsive taxa that reactivate in the immediate aftermath of phytohormone exposure and those that persist and increase over an extended period of 14 days, the latter indicating a potential for microbial stability well beyond the stress exposure. The persistence of stress-responsive taxa over time suggests the use of stress-associated phytohormones or phytohormone-mediated soil priming to enhance growth. Taxa that enrich temporarily may be less effective as bioinoculants, since they likely respond to stress as a signal and are not able to grow and persist over time, thus diminishing their effectiveness during a subsequent stress exposure. Thus, the enrichment and depletion of taxa post-stress are directly tied to the effectiveness of those candidates as bioinoculants. Our study identifies promising targets for exploration in greenhouse and field trials, with direct application to plants during stress, to assess their efficacy.

A limitation of this study is that we did not assess whether microbes utilized the phytohormones ABA and SA. A previous study reported that increasing ABA concentrations led to increased soil respiration, suggesting that microbes utilized ABA as a substrate. However, they also observed a disproportionate increase in respiration relative to the minimal phytohormone inputs of jasmonic acid (JA) and 1-aminocyclopropane-1-carboxylic acid (ACC), suggesting that these instead stimulated microbial mineralization of existing soil carbon (29). Here, we observed a continued persistence of activated taxa in response to phytohormones over 14 days in a few taxa (specifically the log twofold enriched taxa in red—Fig. 4), which could also be an indirect response due to a similar priming effect. However, the immediate activation of bacterial taxa within 24 h suggests that the phytohormones can dually serve as a signal, potentially pointing to a plant's cry for help. We, however, emphasize that we do not interpret this as a direct response to a plant since we did not use plants in the experiment. We used phytohormones as one signal that could be used as a plant's "cry for help," and we are cautious not to extrapolate our findings directly to a specific plant, even though the legacy impact of the plant's assembly of the root zone microbiome remains critical in our findings. Another limitation of this study is that the 16S rRNA and rRNA gene sequencing

method provides a population-level indication of whether a taxon is active (yes or no) and no indication of relative activity levels or individual activity states. Thus, we cannot know whether all individuals within that taxon consistently responded or if some cells remained inactive while others became highly active. There can also be an effect of the extent of rRNA amplification on classification accuracy as cells transition from dormant to growing. *In silico* simulations using a mixture of rRNA amplification models showed up to a possible 47% error rate due to false negatives (e.g., active populations misclassified as dormant [82]), suggesting that active classifications may be more robust than dormant. Furthermore, the life history strategy of a cell may make it difficult to compare relative activities (83) though we do not attempt this here by applying a conservative binary definition per population (active or not). In the future, coupling rRNA:rRNA gene ratio methods with additional approaches such as metabolic capacity measurements can improve understanding of the active fractions of environmental microbiomes (84).

Overall, the results of this experiment show that microbial reactivation can be used to identify both immediate and persistent bacterial responders within the root zone soil microbiome in response to plant stress signals. The actinobacterial members, such as *Microbispora* sp., were consistently noted among the responsive bacterial populations across the different phytohormones and soils. The next steps should investigate the molecular mechanisms and the dynamic host-microbiome feedback with *Microbispora* sp., given phytohormones as stress signals from the plant.

## ACKNOWLEDGMENTS

Support for this research was provided by the United States National Science Foundation under Grant No. MCB #1817377 to A.S. Additional support was provided by the Great Lakes Bioenergy Research Center, U.S. Department of Energy, Office of Science, Office of Biological and Environmental Research under Award Number DE-SC0018409 and by the National Science Foundation Long-Term Ecological Research Program (DEB #1832042). Additional support was provided by the Michigan State University Plant Resilience Institute, the USDA National Institute of Food and Agriculture, and Michigan State University AgBioResearch. A.S. acknowledges finishing project support from the Agriculture and Food Research Initiative (AFRI) of the USDA National Institute of Food and Agriculture (NIFA) Grant number #2024-67019-42477 and from the European Union (ERC, MicroRescue, 101087042). Views and opinions expressed are, however, those of the author(s) only and do not necessarily reflect those of the European Union or the European Research Council. Neither the European Union nor the granting authority can be held responsible for them.

## AUTHOR AFFILIATIONS

[1]The Plant Resilience Institute, Michigan State University, East Lansing, Michigan, USA
[2]The Great Lakes Bioenergy Research Center, Michigan State University, East Lansing, Michigan, USA
[3]Universite Claude Bernard Lyon 1, CNRS, INRAE, VetAgro Sup, Laboratoire d'Ecologie Microbienne LEM, CNRS UMR5557, INRAE UMR1418, Villeurbanne, France

## AUTHOR ORCIDs

Sreejata Bandopadhyay  http://orcid.org/0000-0002-4694-2461
Ashley Shade  http://orcid.org/0000-0002-7189-3067

## FUNDING

| Funder | Grant(s) | Author(s) |
| --- | --- | --- |
| National Science Foundation | 1817377 | Ashley Shade |
| U.S. Department of Energy | DE-SC0018409 | Ashley Shade |

| Funder | Grant(s) | Author(s) |
|---|---|---|
| National Science Foundation | 1832042 | Oishi Bagchi |
| | | Ashley Shade |
| AgBioResearch, Michigan State University | | Ashley Shade |
| U.S. Department of Agriculture | 2024-67019-42477 | Ashley Shade |
| European Research Council | 101087042 | Ashley Shade |

## AUTHOR CONTRIBUTIONS

Sreejata Bandopadhyay, Conceptualization, Data curation, Formal analysis, Investigation, Methodology, Visualization, Writing – original draft, Writing – review and editing | Oishi Bagchi, Data curation, Investigation, Methodology, Writing – review and editing | Ashley Shade, Conceptualization, Data curation, Funding acquisition, Investigation, Methodology, Project administration, Supervision, Visualization, Writing – original draft, Writing – review and editing

## DATA AVAILABILITY

Sequence data have been deposited in the Sequence Read Archive under BioProject PRJNA932434. The scripts to analyze the data and generate figures are on GitHub (https://github.com/ShadeLab/PAPER_Dormancy_resuscitation_phytohormone_mesocosm_Bandopadhyay2025).

## ADDITIONAL FILES

The following material is available online.

### Supplemental Material

**Supplemental material (Spectrum01789-25-s0001.pdf).** Table S1; Fig. S1 to S6.

### Open Peer Review

**PEER REVIEW HISTORY (review-history.pdf).** An accounting of the reviewer comments and feedback.

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
