## [Reviewer comments · Microbiology Spectrum]

Microbiology Spectrum

Activation dynamics and assembly of root zone soil bacterial communities in response to stress-associated phytohormones

Sreejata Bandopadhyay, Oishi Bagchi, and Ashley Shade

Corresponding Author(s): Ashley Shade, CNRS Delegation Alpes

Review Timeline:

Submission Date:	September 1, 2025
Editorial Decision:	September 5, 2025
Revision Received:	September 15, 2025
Accepted:	September 16, 2025

Editor: Blaire Steven

Reviewer(s): The reviewers have opted to remain anonymous.

Transaction Report:

DOI: <https://doi.org/10.1128/spectrum.01789-25>

Re: Spectrum01789-25 (**Activation dynamics and assembly of root zone soil bacterial communities in response to stress-associated phytohormones**)

Dear Prof. Ashley Shade:

Thank you for the privilege of reviewing your work. Below you will find my comments, instructions from the Spectrum editorial office, and the reviewer comments.

Having read the response to the reviews and current version of the manuscript I am happy to report that I think the manuscript is sufficiently revised and is suitable for publication in Microbiology Spectrum.

My one comment on reading the manuscript, is I think the authors have missed some literature that questions the interpretation of 16S transcript/gene ratios as an indicator of activity. I don't think this affects the interpretation of results, but should be acknowledged in the modified re-submission.

I am pleased to inform you that your manuscript has been editorially accepted for publication. However, there are a few additional questions in the submission form that need to be answered before the final decision. Once these are completed, please return your submission so that I can move your paper forward to acceptance.

Revision Guidelines

Sincerely,
Blair Steven

Response to editorial comment

Having read the response to the reviews and current version of the manuscript I am happy to report that I think the manuscript is sufficiently revised and is suitable for publication in *Microbiology Spectrum*.

>>> Thank you for your positive comments and expedited review.

My one comment on reading the manuscript, is I think the authors have missed some literature that questions the interpretation of 16S transcript/gene ratios as an indicator of activity. I don't think this affects the interpretation of results, but should be acknowledged in the modified re-submission.

>>> Thank you for this comment. We have now added three new references and discussion Lines 583-592 in the marked up manuscript file to discuss in more detail the limitations of the use of the 16S rRNA:rRNA gene method.

Added references:

82. Steven B, Hesse C, Soghigian J, Gallegos-Graves LV, Dunbar J. 2017. Simulated rRNA/DNA Ratios Show Potential To Misclassify Active Populations as Dormant. *Applied and Environmental Microbiology* 83:e00696-17.
83. Blazewicz SJ, Barnard RL, Daly RA, Firestone MK. 2013. Evaluating rRNA as an indicator of microbial activity in environmental communities: limitations and uses. *The ISME Journal* 7:2061-2068.
84. Wang Y, Thompson KN, Yan Y, Short MI, Zhang Y, Franzosa EA, Shen J, Hartmann EM, Huttenhower C. 2023. RNA-based amplicon sequencing is ineffective in measuring metabolic activity in environmental microbial communities. *Microbiome* 11:131.

Re: Spectrum01789-25R1 (**Activation dynamics and assembly of root zone soil bacterial communities in response to stress-associated phytohormones**)

Dear Prof. Ashley Shade:

Thank you for providing your edits, your manuscript has been accepted, and I am forwarding it to the ASM production staff for publication. Your paper will first be checked to make sure all elements meet the technical requirements. ASM staff will contact you if anything needs to be revised before copyediting and production can begin. Otherwise, you will be notified when your proofs are ready to be viewed.

Sincerely,
Blair Steven
Editor
Microbiology Spectrum